# RPDNet: Automatic Fabric Defect Detection Based on a Convolutional Neural Network and Repeated Pattern Analysis

**DOI:** 10.3390/s22166226

**Published:** 2022-08-19

**Authors:** Yubo Huang, Zhong Xiang

**Affiliations:** Faculty of Mechanical Engineering and Automation, Zhejiang Sci-Tech University, Hangzhou 310018, China

**Keywords:** fabric defect detection, convolutional neural network, repeated pattern analysis

## Abstract

On a global scale, the process of automatic defect detection represents a critical stage of quality control in textile industries. In this paper, a semantic segmentation network using a repeated pattern analysis algorithm is proposed for pixel-level detection of fabric defects, which is termed RPDNet (repeated pattern defect network). Specifically, we utilize a repeated pattern detector based on convolutional neural network (CNN) to detect periodic patterns in fabric images. Through the acquired repeated pattern information and proper guidance of the network in a high-level semantic space, the ability to understand periodic feature knowledge and emphasize potential defect areas is realized. Concurrently, we propose a semi-supervised learning scheme to inject the periodic knowledge into the model separately, which enables the model to function independently from further pre-calculation during detection, so there is no additional network capacity required and no loss in detection speed caused. In addition, the model integrates two advanced architectures of DeeplabV3+ and GhostNet to effectively implement lightweight fabric defect detection. The comparative experiments on repeated pattern fabric images highlights the potential of the algorithm to determine competitive detection results without incurring further computational cost.

## 1. Introduction

During the process of textile production, discernible defects with varying differences in physical size and shape inevitably reside on the fabric surface owing to factors such as machine failure, damage, and pollution, thereby adversely impacting the price realization ability of finished products [1]. While manual detection can be performed to identify non-conforming products, inadequate attention and poor visual judgment of workers proves to be unreliable, with an accuracy of only 60–75% [1]. In order to overcome such problems, ingenious algorithms and techniques have been developed in recent decades to automate the detection of fabrics with impressive results [2,3]. In the context of the modern textile industry, automatic detection technology pertaining to fabric defects present a recurring challenge in the pursuit of efficiency, amidst growing concerns from the industry and academia.

According to the feature extraction strategy, the automatic defect detection method can be categorized into two strategies—the traditional method and deep learning method. Traditional methods are typically based on artificially designed features aimed at representing texture defects. In particular, it can be divided into four sub-categories [4], namely spectral, statistical, structural, and model-based approaches. While these traditional methods can achieve satisfactory results when applied to specific categories of products, modern production mandates the handling of complex and varied image texture patterns, in conjunction with the assistance of suitable hardware equipment and production environment. In general, traditional methods are relatively difficult to generalize in new operating environments and integrate into new designs in most cases. In recent years, many defect detection networks have been developed to leverage the properties of deep neural networks to improve overall performance [5,6,7,8,9]. Deep learning models can automatically learn useful features from artificially constructed datasets. Subsequently, these acquired features possess stronger expressive power in comparison to artificially designed features. In the event that new fabric design schemes are required or the sensor acquisition environment is altered, the defect detection network can be easily redeployed.

Globally, large-scale industrial production constitutes a major application domain of modern fabric defect detection, where similar equipment procedures can result in the formation of repeated patterns on the product surface. Since reference to similar patterns in the algorithm can help identify potential anomalies, notable advancements in automatic fabric detection have considered analytical methods on periodic patterns. Specifically, Ngan et al. proposed a motif-based defect detection method and included a preprocessing step of lattice extraction, which segmented the fabric image into non-overlapping basic units through a computational model [10]. Jing et al. leveraged the distance matching function to decompose the fabric image of the smooth pattern into small pieces and utilized the CNN model of transfer learning to slide on the whole image to identify defects [11]. In addition, Jia et al. proposed a variety of defect detection methods based on lattice segmentation [12,13,14], and the core of these works is to turn fabric defect detection into a problem of estimating lattice similarity. However, the image periodic pattern analysis based on computational methods usually requires high stationarity and low complexity of the pattern period. Moreover, an insolvable dilemma concerning these works pertains to communication failure between parts of the image due to lattice segmentation. Thus, relying on the comparison between small parts may neglect the abnormality of the image as a whole.

Based on the aforementioned problems, this paper proposes an automatic fabric detection method contingent on the semantic segmentation network and repeated pattern analysis algorithm. Specifically, the implementation of our method does not pass the comparison between basic patterns, but obtains repeated pattern information through pre-computation, and uses it to guide the network in the high-level semantic space to understand repeated feature knowledge and highlight potential defect areas. The considered repeated pattern detector [15] is a CNN-based method that exploits feature activations in the network to detect spatial repeating patterns, which can more accurately capture high-level contextual semantics and extract periodic primitives. However, for our method, injecting the acquired repeated pattern information into the network in the form of image labels requires a great deal of pre-computation, for which we try to replace the whole tedious pre-processing with semi-supervised learning of the network. This semi-supervised scheme uses a set of fused semantic segmentation networks, with shared parameters among the networks, and the original image and corresponding repeated pattern labels are input into different network spaces. During the gradient iteration process, the network group needs to simultaneously achieve semantic segmentation of fabric images and repeated pattern labels. After the conclusion of training, only the necessary parts of the entire network are retained, thereby ensuring that the model does not rely on additional inputs and understands potential repeated pattern knowledge. Concurrently, our method emphasizes the applicability of lightweight detection tasks and integrates two advanced architectures of DeeplabV3+ [16] and GhostNet [17]. The main body is based on the architecture DeeplabV3+ while the lightweight network GhostNet is adopted as the backbone network to enhance overall performance. Furthermore, the proposed algorithm constitutes a general strategy with good generalization ability and can be applied to a variety of mainstream semantic segmentation architectures.

The following sections of this paper are organized in this manner—Section 2 introduces the relevant work of repeated pattern analysis algorithm and automatic defect detection algorithm. In Section 3, the procedure of the proposed RPDNet is elaborated and discussed. In Section 4, the results of several qualitative and quantitative experiments are reported. Finally, Section 5 includes the conclusion of this work.

## 2. Related Works

In this section, we briefly introduce and discuss related works on image periodic pattern analysis and automatic defect detection algorithms.

### 2.1. Periodic Pattern Analysis

Periodic patterns of patterns are ubiquitous in reality, and they can provide strong clues to the geometric features of the elements that make up the structure, which are beneficial to visual detection algorithms. For decades, researchers have proposed many methods to analyze the periodicity of repeated patterns. The gray level co-occurrence matrix (GLCM)-based method is one of the most typical methods [18] with being time-consuming. In order to reduce computational requirements, Unser proposed the sum and difference histograms method as an alternative to GLCM [19]. In more modern approaches, Liu et al. proposed a computational model for periodic pattern analysis based on crystallographic group theory [20]. Nasri et al. sensed repeated patterns by detecting peak correlations in images with an autocorrelation function (ACF), and genetic algorithm for optimization [21]. Xiang et al. proposed a method based on adaptive template matching to find repeated patterns with good flexibility [22]. While these periodic analysis methods have been shown to be effective on some types of images, only low-level features of the image are usually used. With the development of deep learning, a recent study started to use CNN to implement a more robust repeated pattern detector [15]. This method utilizes activations produced by convolutional filters to infer texture placement rules and seek out repeated patterns, which captures the high-level contextual semantics of an image well and is robust to the appearance and geometric variations of anomalous regions in an image. Due to the superiority of neural network, we introduce this method in this work.

### 2.2. Automatic Defect Detection

In the development of automatic defect detection techniques, the majority of the traditional methods utilize artificially designed features to distinguish between defective and non-defective areas. Mallik-Goswami et al. used a strictly selected structural element to conduct morphological operations on spatially filtered defect images of fabrics [23]. Tsai et al. proposed a global method based on Fourier transform with curvature analysis to filter the repeated patterns of images and isolate the retained defects from it [24]. Cohen et al. used a Gaussian Markov random field-based approach to model defect-free fabrics and detect defects by comparison with test images [25]. Haiqin et al. proposed to use texture enhancement method to improve the discrimination between texture background and defect area, and combined with GLCM algorithm to extract features for classification [26].

As machine learning becomes mainstream in multiple vision domains [27,28,29] in recent years, many researchers have utilized CNNs to design more robust and faster defect detection networks. Ren et al. extracted features on image patches through CNN, and then located defect regions through a pixel-level prediction module [30]. Li et al. proposed a compact network architecture for fabric defect detection, which uses a variety of network optimization techniques to reduce model size and improve the accuracy of real-time detection [31]. Jing et al. proposed a pixel-level defect detection method using a combination of pre-trained MobileNetV2 and UNet [32]. Another family of defect detection methods in deep learning is the method based on generative models. Such methods are unsupervised, and the core idea is to train a model with flawless images to reconstruct test images without relying on the tedious data labeling process. For example, Hu et al. reconstructed a given query image through a generative adversarial network and discovered potential defect regions by creating a residual map with the original image [33]. Mei et al. proposed an unsupervised automated method to detect and localize defects in texture through a multi-scale convolutional denoising autoencoder architecture and Gaussian pyramid [34]. Hu et al. used convolutional denoising auto-encoder and hash encoder to extract and retrieve local patterns of printed fabrics, and completed defect localization by processing the difference mapping between images [35].

Although deep learning-based defect detection methods have the ability to handle high-level semantic context, these methods are built on the basis of natural image vision, which is not designed for images containing repeated textures. The following problem shows that in the process of finding defect regions from similar patterns, there is no explicit guidance for the network to understand those periodic feature knowledge. Intuitively, we believe that injecting latent periodic information into the network is helpful for identifying anomalies in textures, so a deep learning-based pixel-level defect detection method is specially designed for fabric images containing periodic textures.

## 3. Methodology

We propose an automatic defect detection method for fabrics based on repeated pattern analysis and semantic segmentation algorithm. It is an efficient end-to-end image segmentation framework. In this section, we will describe specific implementation steps of the method in detail, and the steps are divided into the repeated pattern analysis stage and semantic segmentation stage.

### 3.1. Repeated Pattern Analysis Stage

A CNN can be considered as a set of feature extractors $f_{l}$ with different levels of abstraction, through which regular local peaks $p\in P_{f_{l}}$ are generated in each layer of the network. An example is shown in Figure 1.

By following the method of Lettry et al. [15], we obtain the activation peaks of each layer of the network through AlexNet [36] with the same configuration. Each group of peaks $\left(p^{m},p^{n}\right)$ corresponds to a set of displacement vectors $d^{m,n}$ and forms a set $D_{f_{l}}$, in which the displacement vector with the highest probability is most likely the size of the repeated pattern.
$$D_{f_{l}}=\left\{d^{m,n}:\left|p^{m}-p^{n}\right|,\forall p^{m},p^{n}\in P_{f_{l}},m\neq n\right\}$$

The displacement vectors dm,n∈Dfl of all filters fl∈Fl and layers l∈L are then voted into the Hough voting space H:ℝ2→ℝ. Due to different feature layers, each vote obeys a two-dimensional normal distribution centered on dm,n with σl. At the same time, since different filters may lead to different numbers of activation peaks, the votes are weighted based on the number of vectors in each set Dfl.
(1)H=∑x∈ℝ2∑l∈Lfl∈Fl1Dfl∑dm,n∈DflHfl,m,n
where
$$H_{f_{l},m,n}=\frac{1}{2\pi \sqrt{\left|\Sigma \right|}}\exp\left(-\frac{1}{2}{\left(x-d^{m,n}\right)}^{⊤}\Sigma^{-1}\left(x-d^{m,n}\right)\right)$$
$$\Sigma =\left(\begin{array}{cc} \sigma_{l}^{2} & 0\\ 0 & \sigma_{l}^{2} \end{array}\right)$$

Assume an axially aligned rectangular grid on the x-y axes to obtain the most suitable displacement vector $d^{*}$, which is the size of the repeated pattern, the calculation formula is as follows:(2)$$d^{*}=\left(\underset{x}{\mathrm{argmax}}\;H_{x,0},\underset{y}{\mathrm{argmax}}\;H_{0,y}\right)$$

After getting the size of the repeated pattern, it is necessary to further segment all repeated patterns in the fabric image. First, by calculating the voting consistency weight $w_{f_{l}}$, the filter $F_{l}{}^{*}$ with better consistency with the selected displacement vector $d^{*}$ is selected.
(3)$$w_{f_{l}}=\sum_{d^{m,n}\in D_{f_{l}}^{*}}\left[\frac{1}{\left|D_{f_{l}}\right|+\phi }\cdot \exp\left(-\frac{{\left\|d^{m,n}-d^{*}\right\|}^{2}}{2\alpha_{l}^{2}}\right)\right]$$
where $D_{f_{l}}^{*}=\left\{d^{m,n}\in D_{f_{l}}:\left\|d^{m,n}-d^{*}\right\|<3\alpha_{l}\right\}$ is the total number of votes for which the displacement vector $d^{m,n}$ is consistent with $d^{*}$. $\alpha_{l}^{}$ is the neighborhood radius considered at layer l, ϕ is a prior estimate, both of which adopted the reference configuration [15]. We keep filters with weights greater than $\delta w_{f_{l}}^{*}$, where $w_{f_{l}}^{*}$ is the maximum weight achieved. Since it is easier for the filters to produce suitable outputs for the fabric samples under consideration, we set δ to 0.8 for more reliable results.

After that, we vote for the texture’s centroids via the implicit pattern model (IPM). To begin with, in order to obtain the relative positions of the displacement vector votes, a simplification process is required in the modulo space:M:ℝ2→[0,dx*]×[0,dy*]
M(v)→vxmoddx*,vymoddy*

Then calculate the offset $o^{*}=(o_{x},o_{y})$ to minimize the distance of the consistent votes of the filters $F_{l}{}^{*}$ to the center of the pattern:(4)$$o^{*}=\underset{o}{\mathrm{argmin}}\sum_{\begin{array}{c} l\in L\\ c_{m}\in F_{l}^{*}\\ d^{m,n}\in D_{f_{l}} \end{array}}w_{m,n,f_{l}^{*}}\left\|M\left(d^{m,n}-o\right)-d^{*}/2\right\|$$

Finally, the IPM is used to detect fabric instance, on which a model of a two-dimensional grid layout is fitted. After getting the grid layout, we perform related post-processing, with an aim to convert the results of repeated pattern detection to the form of labels and then use them in the semantic segmentation network. This process is shown in Figure 2.

Here we assume that the pattern periods of the fabric images under consideration are stationary, so it is expected that the labels obtained should contain all parts of the images. But due to the lack of information at edges and defects, these places are usually not responded by the selected filters (Figure 2a). For this reason, we fill the missing areas with the grid layout and repeated pattern size that have been obtained, so that the repeated texture grids will be tiled across the entire image (Figure 2b). The response of the convolutional filters capture the non-rigid micro-deformation of the fabric to some extent. We divide the adjacent grids into two groups for labeling to better utilize the spatial distortion information captured by the elastic grids, and obtain input labels to inject into the network space of semantic segmentation (Figure 2c). At the same time, we make a copy of these labels and mark the defect regions, which serve as target labels for the network (Figure 2d). Obviously, the input labels contain explicit repeated texture information, and in the target labels, the grids that contain defect regions are different from those that do not in terms of texture. In the next step, we intend to incorporate periodic patterns knowledge of fabric images into a semantic segmentation model, with a view to adaptively improve its defect recognition ability.

### 3.2. Semantic Segmentation Stage

In supervised learning of semantic segmentation, additional input knowledge and target knowledge often lead to more effective results. Based on this, we exploit the periodic patterns knowledge of fabric images through two aspects, one is to highlight the potentially defective regions by distinguishing abnormal patches in the target labels of the network, and the other is to input the information of the repeated pattern into the network space to exploit the potential similarity of high-level semantic features. The former can be implemented by an additional pixel-level module and a cross-entropy loss to output semantically segmented images containing potentially anomalous regions, while the latter by injecting repeated pattern informative labels and additional convolution computations in the network.

However, as previously described, the calculation process of repeated pattern analysis is quite cumbersome. If we directly inject the label input and the image input into the same network space, it will cause the method to rely on the calculation of labels, which will seriously affect the detection speed. We therefore consider a semi-supervised scheme, which intends to remove this reliance by introducing a pair of additional repeated pattern segmentation networks to feed the repeated pattern labels into the architecture independently.

Our model encompasses a set of semantic segmentation networks that blend two advanced architectures, i.e., DeeplabV3+ and GhostNet. Figure 3 shows the proposed architecture containing three encoder-decoder networks to fully exploit the periodic pattern knowledge of fabric images achieved by the repeated pattern analysis algorithm.

We denote the encoder as E, and the decoder as D. According to the features extracted from GhostNet, the communication between E1 to D2 and E1 to D3 is established through skip connection, thus forming several combined networks. The main network E1-D1-D2-D3 receives the original sample input, and the output semantic segmentation results include fabric defect images and repeated pattern images that contain potential abnormal areas. While E1-E2-D2 and E1-E3-D3 are repeated pattern semantic segmentation networks, where two independent E networks are used to receive fabric images and corresponding repeated pattern input labels respectively. In the training stage, the gradient of each node in the entire network is updated at the same time. When the training is completed, we discard unnecessary model parts. In this case, the remaining part (red dotted box in Figure 3) is able to detect defects in the fabric images independently.

We introduce DeeplabV3+ [16] as the basic framework of semantic segmentation network, which is an encoder-decoder network focusing on efficiency (Figure 4). In the encoder of DeeplabV3+, low-level features and deep-level features are obtained through the backbone network at the same time, and then the deep-level features are enhanced with the Atrous Spatial Pyramid Pooling (ASPP) module [27], which is a deep feature extraction module based on multiple parallel atrous convolution with different rates. Next, after 1 × 1 convolution and bilinear upsampling operation, low-level features and deep-level enhancement features are merged in the decoder, and then the semantic segmentation results are output through 3 × 3 convolution and bilinear upsampling operation.

We use GhostNet [17] with the head classification module removed as the backbone network of DeeplabV3+. GhostNet is an advanced deep learning architecture designed for embedded devices, which emphasizes reducing the computational consumption of the process of generating redundant feature maps. Its core is an efficient convolution scheme based on Ghost modules. Different from general efficient scheme that use pointwise convolution and depthwise convolution to process cross-channel features and spatial information in turn [37], Ghost module first employs ordinary convolution, on top of which it takes an inexpensive linear operation, and stacks the results of the two on the channel, which contains both channel features and spatial information. Through residual connection and two Ghost modules, Ghost Bottleneck (G-bneck), the basic unit of GhostNet, can be formed (as shown in Figure 5).

GhostNet surpasses the performance of MobileNetV3 [38] on the ImageNet ILSVRC2012 classification dataset, which can effectively improve the applicability of lightweight CNN architecture. Table 1 shows the details of the GhostNet we used.

### 3.3. Training Procedure

The most classic way to train a semantic segmentation network is to employ the cross-entropy (CE) loss, where *W* and *H* represent the width and height of image, respectively, and $p_{i,j}$ is the probability of the defect class at the position (*i*, *j*) estimated by the network.
(5)$$L=\sum_{i=1}^{W}\sum_{j=1}^{H}p_{i,j}\log\left(p_{i,j}\right)$$

While for fabric defect detection, the number of defects and background pixels is highly unbalanced, and the loss of the simple class can overwhelm the loss of the rare class through the use of cross-entropy loss. In this regard, we introduce focal loss to alleviate this problem [39].
(6)$$L_{Focal}=\sum_{i=1}^{W}\sum_{j=1}^{H}{\left(1-p_{i,j}\right)}^{\gamma }\log\left(p_{i,j}\right)$$
where ${\left(1-p_{i,j}\right)}^{\gamma }$ is the dynamic adjustment factor. When the predicted probability is high, the dynamic adjustment factor approaches 0, while when the predicted probability is low, it approaches 1, so as to achieve the effect of mining difficult samples. Empirically, we set γ = 2.

The proposed RPDNet is implemented based on the PyTorch [40] framework. All training is performed on a server with four GeForce RTX 2080 Tis. The stochastic gradient descent optimizer [41] is used with a momentum of 0.9 and a weight decay parameter of 0.0001. The batch size is 8. The learning rate is set to 0.005. The number of iterations of the network is usually 10 k, and the parameters of E2-D2 and E3-D3 are frozen at a quarter of the number of iterations to speed up training.

## 4. Experiments and Discussion

This section describes a series of experiments to evaluate the performance of RPDNet. These experiments involve two fabric image benchmark databases: Fabric Images database (FI) [42] and TILDA Textile Texture Database [43]. The FI dataset has 106 images including star, dot, and box fabric patterns, and is divided into six defect types: broken end, hole, knot, netting multiple, thick bar, and thin bar. The TILDA database includes eight common types of textiles. Three types of fabric images containing repeated textures are selected and divided into four defect types: hole, color spot, thread error, and foreign body, with a total of 750 images. All images are strictly manually labeled for each pixel by one-hot encoding, as Figure 6 shows some representative examples.

We perform data augmentation by cropping and horizontal flipping, and resize the image to 256^2^. We set training and test sets in a 3:1 ratio for all images and shuffle the order. Table 2 provides the specific details of the datasets. With four GeForce RTX 2080 Tis, the training time is about 6 h for the FI dataset and about 10 h for the TILDA dataset.

### 4.1. Performance Evaluation

With a view to comprehensively verify the defect detection performance of the proposed method, the detection results of the traditional method LSTS [14] and four deep learning-based supervised algorithms, including FCN [44], UNet [45], Mobile-Unet [32], and RPDNet are qualitatively and quantitatively compared. Among them, LSTS is a traditional fabric detection method based on lattice segmentation and template statistics. The method infers the placement rule of texture primitives based on “peaks” found on rows and columns of thresholded images, and then identifies defective lattices through multiple feature extraction methods and template statistics. We default to using the most efficient combination of feature extraction methods and associated parameter settings. In particular, UNet and FCN are popular deep learning semantic segmentation models that have been extensively adopted in image segmentation tasks in various domains with excellent results [46,47]. Likewise, Mobile-Unet is a deep learning model developed in the context of fabric defect detection which combines the advanced lightweight architecture MobileNetV2 on the basis of UNet. Furthermore, all comparisons are based on the same conditions.

Subsequently, four evaluation metrics are adopted: Intersection over Union (IoU), Recall, Precision, and F1-Measure, which are frequently used in image defect detection. These metrics are defined as follows:(7)$$IoU=\frac{TP_{p}}{FN_{p}+TP_{p}+FP_{p}}$$
(8)$$Recall=\frac{TP_{p}}{FN_{p}+TP_{p}}$$
(9)$$Precision=\frac{TP_{p}}{FP_{p}+TP_{p}}$$
(10)$$F1-Measure=\frac{2\cdot Precision\;\cdot \;Recall\;}{\;Precision+Recall\;}$$

As shown in Figure 7, *FP*_P_ represents the wrongly segmented defect area in the background, *TP*_P_ represents the correctly segmented defect area in the ground truth, and *FN*_P_ represents the undetected area in the ground truth. For the segmentation methods of deep learning, we divide each pixel of the image into the corresponding types of the above areas in conformity with the predicted defect area and ground truth. For LSTS, its result is a region of lattices, so according to Jia et al. [14], the following guidelines are used to transform: retain any correct pixels if the detection lattice contains them, otherwise set the region of the detection lattice to foreground to indicate an error.

According to the aforementioned evaluation metrics, we record the quantitative comparison results in Table 3. Evidently, the results highlight achieving the best comprehensive detection performance on both the FI dataset and the TILDA dataset by the proposed RPDNet. On average, RPDNet outperforms the second-best method by a significant margin of 7.9%, 5.7%, 2.9%, and 4.4% in IoU, Recall, Precision, and F1-Measure on both datasets, respectively, which demonstrates the excellent performance of RPDNet in fabric defect detection. It is also worth noting that the various methods generally perform better on the TILDA dataset than on the FI dataset, because the FI dataset has more types of defects and has fewer samples than the TILDA dataset, which makes the network learning is more difficult on the FI dataset.

On a GTX 2080Ti GPU, the average detection time of the proposed RPDNet is 6.59 ms, exceeding 150FPS. Therefore, RPDNet is capable of meeting the requirements of real-time detection for practical industrial applications.

According to Figure 8, the pixel locations and their classes of images output by RPDNet are almost comparable to ground true images. In comparison to other three deep learning methods, our method possesses the contour closest to the target from a visual standpoint. In particular, FCN and UNet are the most classic image segmentation networks, both of which utilize a fully convolutional architecture for advanced feature extraction and fusion. In contrast with FCN, UNet adopts a symmetric encoder-decoder structure to capture clearer object boundaries through gradual restoration of spatial information. From our experimental results, the lack of noticeable discrepancies in the performance of FCN and UNet in fabric defect detection is evident. Despite both networks being capable of segmenting the defects of the samples, there exists a specific error in the class of output pixels. Additionally, Mobile-Unet is based on a pre-trained MobileNetV2 network, which integrates advanced neural network optimization techniques and boasts superior performance compared to FCN and UNet. As shown in Figure 8e, Mobile-Unet is more sensitive to the texture details of fabric defects, and the class error of output pixels is significantly improved. In view of the strict lattice segmentation process in the case of LSTS [13], the categories of images that can be effectively applied are limited (Figure 8f). In the case of specific or most samples of these image types, peak distribution rules are difficult to be captured in rows and columns owing to insufficient contrast of texture primitives, changes in shooting angle, or non-rigid deformation of fabrics. Furthermore, the lattice segmentation process of LSTS mandates additional computation, while our method utilizes repeated patterns information through self-learning of the network, without affecting the detection efficiency.

### 4.2. Ablation Studies

With the aim of ascertaining the effectiveness of our method, ablation experiments are performed on the TILDA dataset. Subsequently, we perform two types of ablation—removing and reducing the injection of repeated pattern knowledge in RPDNet, respectively. In Table 4, the RPDNet without E2-D2 and E3-D3 of the first group represents the necessary model part for the segmentation task and functions as a reference baseline. In addition, the second group merely utilizes the target labels of repeated pattern and inputs blank labels into E2-D2 and E3-D3. By comparing all the results, it is evident that the evaluation results of the full model are effectively improved by 8.3% on IoU, 4.2% on Recall, 4.9% on Precision, and 4.5% on F1-Measure compared to the first group lacking any repeated pattern knowledge. Ideally, such a degree of improvement demonstrates that specified additional and valid repeated pattern information or reference of the approximate location of defects can better guide model learning and lead the model to find anomalies more easily. In the second experiment comprising blank input labels, the evaluation results are still significantly higher than the reference baseline model and lower than the full model. Such a precedent indicates that the knowledge of target labels positively contributes to the detection ability of RPDNet, thereby yielding more accurate evaluation results through the injected repetitive pattern knowledge. Therefore, these additional network components in RPDNet are able to elevate the overall performance.

### 4.3. Model Extension

The proposed method illustrates a general strategy applicable to various mainstream semantic segmentation architectures. For the purpose of substantiating the generalization of the model, we extend RPDNet by UNet [45], MobileNetv2 [48], and GhostNet [17]. In the case of RPDNet-UNet, we transfer all levels of features from shallow to deep through skip connections between the encoder network and the decoder network. For MobileNetv2 and GhostNet, they are combined as the backbone network for feature extraction under the DeeplabV3+ architecture. We conduct evaluation on the TILDA dataset.

As shown in Table 5, the experimental results of the benchmark models and the extended models are recorded, and RPDNet can effectively improve the evaluation results on all three extensions, which indicates excellent generalization capabilities. Notably, it is observed that the improvement of evaluation metrics in both extended experiments based on DeeplabV3+ is higher than that of RPDNet-UNet. We infer that this is because DeeplabV3+ is an asymmetric encoder-decoder architecture: Since our strategy permits network interaction during the training process, the learning experience becomes easily transferable if the part of the network where the parameters are shared possesses a larger capacity.

### 4.4. Experiments on Repeated-Pattern-Free Images

In numerous instances, trained RPDNet successfully elevates its suitability for the defect detection of repeated pattern fabrics with zero computational increase. In this section, we conduct further studies on repeated-pattern-free images to explore the application object expansion of our method.

The considered repeated pattern detector can detect well-aligned repeating elements and even some potentially repeated visual appearances, and its effectiveness has been demonstrated on those challenging real-world images [15]. In our application scenario, the programmed production of fabrics and the controlled image acquisition environment make it easier for the repetition of images to be captured, which makes this repeated pattern analysis method more applicable. However, despite the CNN’s ability to provide advanced feature processing capabilities, we found that it is still unable to handle images with non-repeating patterns. We believe this is because smaller pattern repetitions are less likely to penetrate all layers from shallow to deep than larger repetitions, thus being difficult to be captured by the network and resulting in unstable peak response trends. This is especially common in very smooth textured test images. In Figure 9, we depict a success–failure example of peak activations for a set of feature maps.

We examine three types of repeated-pattern-free images in the TILDA database (Figure 10). Refer to Section 4 for the processing of the dataset. Since on all three types of images, the CNN repeated pattern detector fails to produce effective peak responses, we try to directly circumvent this limitation to implement our method. Specifically, we directly divide these images into a grid layout of 16^2^ size as input labels (the grid size is an empirical estimate that can accommodate stable textures), and mark the grids containing defects as the target labels of the network.

Subsequently, we conduct two comparative experiments with the baseline reference model and the model with blank input labels. Accordingly, the results are illustrated in Table 6. It is evident that the full model showcases minor improvement compared to the baseline reference model, but also exhibits similar levels of improvement compared to the blank input labels group. In this regard, we believe that these positive contributions should primarily stem from the marked defect regions of interest, while manual segmentation information does not provide effective help to the network. For our approach, we believe that the essence is to use additional knowledge of repeated patterns to increase the priority of those filters that can respond to potential repeated patterns, leading to the instructive significance of exploiting adjacent repeating patterns. On further observation, regular repeated activation responses are not evidently found in the semantic segmentation network, so a nonsensical prioritization tendency in the network is likely to be the direct cause of the failure. Overall, our method fails to adequately handle repeated-pattern-free images. In the future, extending the applicable range of this method will be of paramount importance.

## 5. Conclusions

To summarize, this paper introduces RPDNet; a defect segmentation network for repeated pattern fabric images and has been evaluated on the FI and TILDA datasets. Notably, the trained RPDNet can adequately elevate the applicability of detection and does not require additional computation. Subsequently, we combine two state-of-the-art deep learning architectures, DeepLabV3+ and GhostNet, to achieve competitive results on IoU, Recall, Precision, and F1-Measure. Moreover, the proposed RPDNet is also a general strategy with distinct generalization capabilities, and its effectiveness has been verified on both UNet and DeepLabV3+. Finally, this method holds tremendous potential in the field of computer vision for the extension of various architectures in semantic segmentation.

## Figures and Tables

**Figure 1 sensors-22-06226-f001:**
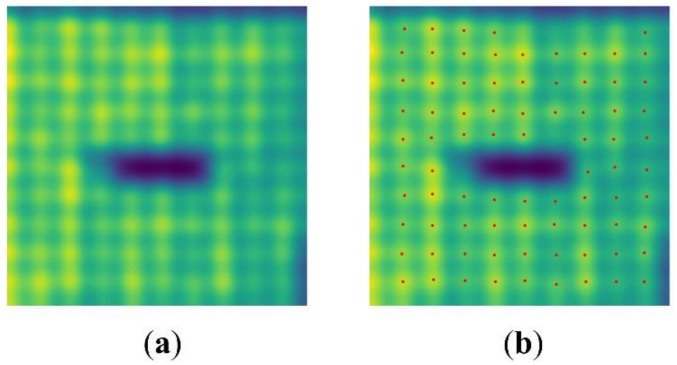
A feature map produced by CNN (**a**); extracted activation peaks (**b**).

**Figure 2 sensors-22-06226-f002:**
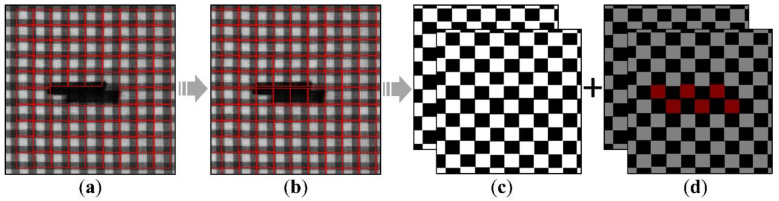
(**a**) The grid layout extracted by the repeated pattern detector and drawn on the original image. (**b**) The edges and defects are filled with grids. (**c**,**d**) The input labels and the one-hot encoded target labels of the semantic segmentation network.

**Figure 3 sensors-22-06226-f003:**
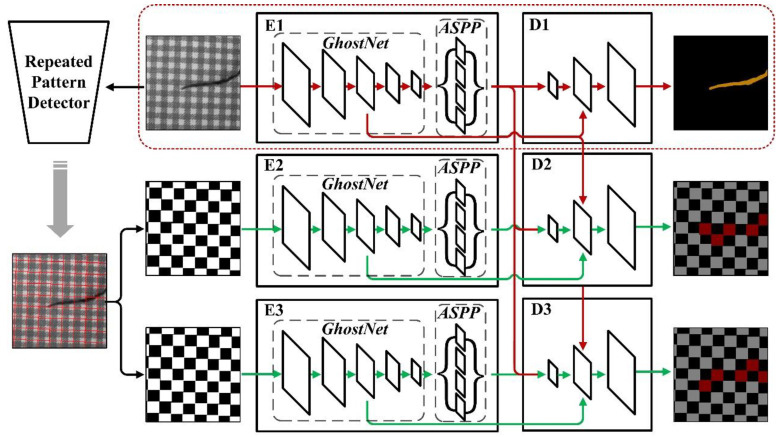
The process of our method. The red dotted box represents the model at the recognition stage. The model allows for network interaction during training by establishing information communication among a set of parallel DeeplabV3+ networks. E and D represent the encoder network and decoder network of DeeplabV3+, respectively.

**Figure 4 sensors-22-06226-f004:**
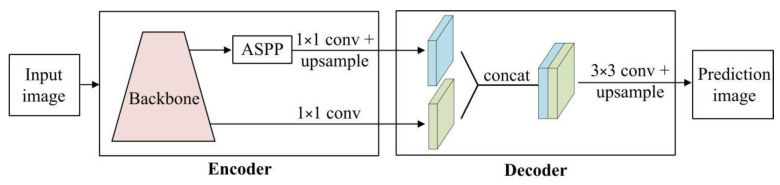
Architecture of DeeplabV3+.

**Figure 5 sensors-22-06226-f005:**
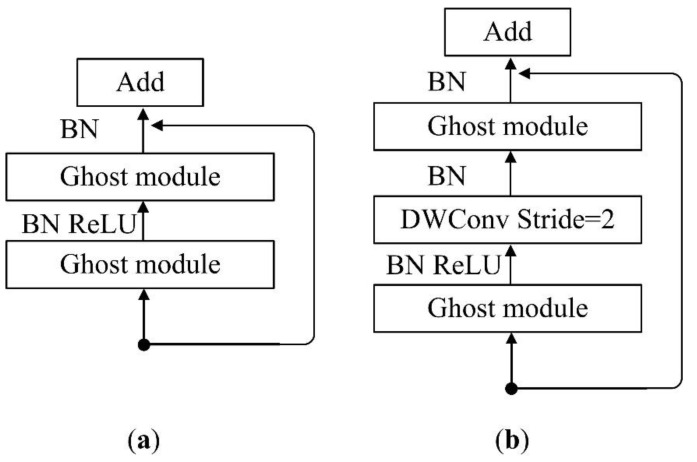
Ghost Bottleneck with stride = 1 (**a**) and 2 (**b**).

**Figure 6 sensors-22-06226-f006:**
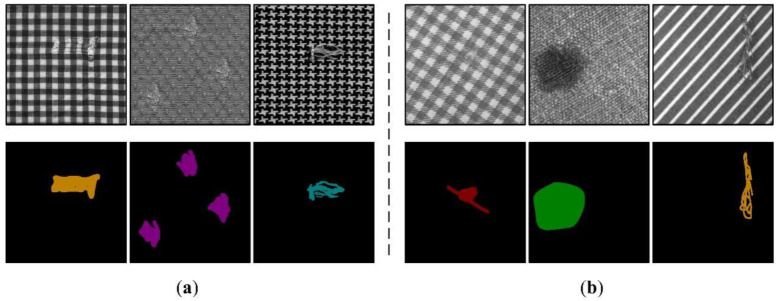
Some representative fabric images and their semantic labels, from (**a**) Fabric Images database, (**b**) TILDA database, respectively.

**Figure 7 sensors-22-06226-f007:**
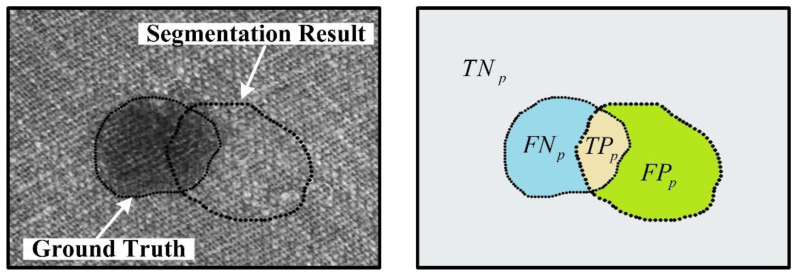
Definition of the *FP*_P_, *TP*_P_, *FN*_P_ and *TN*_P_.

**Figure 8 sensors-22-06226-f008:**
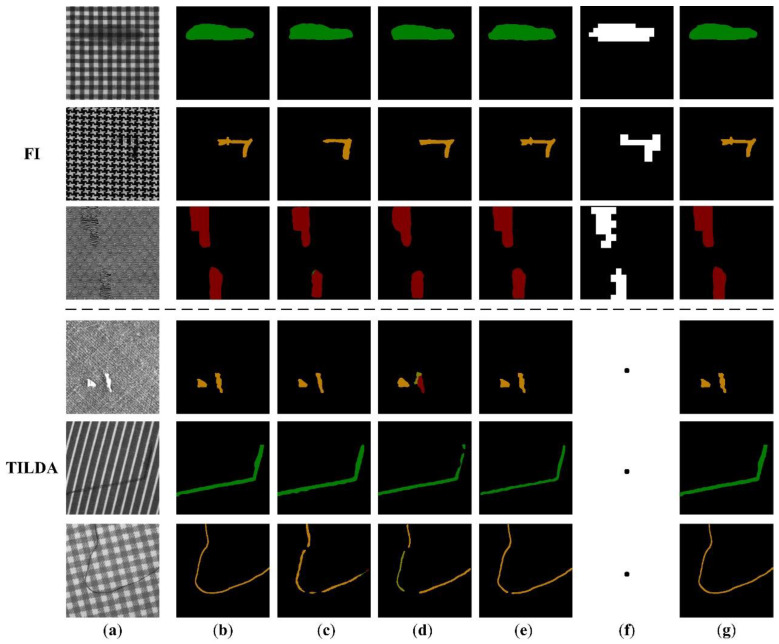
Visual comparison of different detection methods on different fabric samples. (**a**) Original image, (**b**) Ground truth, (**c**) FCN, (**d**) UNet, (**e**) Mobile-Unet, (**f**) LSTS, (**g**) RPDNet.

**Figure 9 sensors-22-06226-f009:**
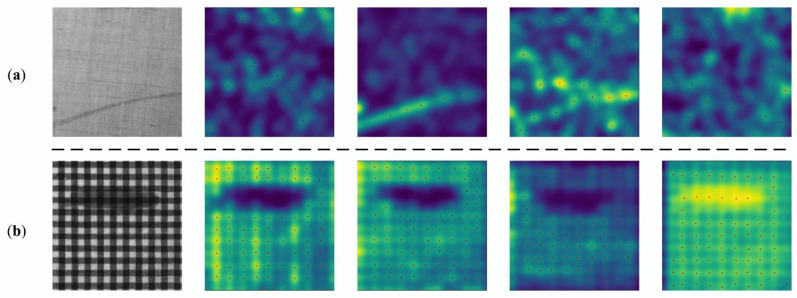
Peak activation of feature maps in random layers in the repeated pattern analysis network. (**a**) is from a repeated-pattern-free fabric image, (**b**) is the contrast response result of a repeated-pattern image.

**Figure 10 sensors-22-06226-f010:**
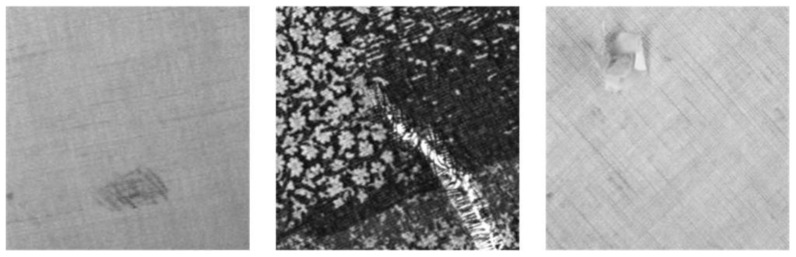
Three representative repeated pattern-free fabric images from the TILDA database.

**Table 1 sensors-22-06226-t001:** Details of GhostNet used. The kernel size is applied to all convolutions in the corresponding G-bneck. #mid represents the number of extended channels in the Ghost module. #out represents the number of output channels. SE stands for a plug-and-play attention module.

Input	Operator	Kernel Size	#Mid	#Out	SE	Stride
256 × 256 × 3	Conv2d	3 × 3	-	16	-	2
128 × 128 × 16	G-bneck	3 × 3	16	16	-	1
128 × 128 × 16	G-bneck	3 × 3	48	24	-	2
64 × 64 × 24	G-bneck	3 × 3	72	24	-	1
64 × 64 × 24	G-bneck	5 × 5	72	40	True	2
32 × 32 × 40	G-bneck	5 × 5	120	40	True	1
32 × 32 × 40	G-bneck	3 × 3	240	80	-	2
16 × 16 × 80	G-bneck	3 × 3	200	80	-	1
16 × 16 × 80	G-bneck	3 × 3	184	80	-	1
16 × 16 × 80	G-bneck	3 × 3	184	80	-	1
16 × 16 × 80	G-bneck	3 × 3	480	112	True	1
16 × 16 × 112	G-bneck	3 × 3	672	112	True	1
16 × 16 × 112	G-bneck	5 × 5	672	160	True	2
8 × 8 × 160	G-bneck	5 × 5	960	160	-	1
8 × 8 × 160	G-bneck	5 × 5	960	160	True	1
8 × 8 × 160	G-bneck	5 × 5	960	160	-	1
8 × 8 × 160	G-bneck	5 × 5	960	160	True	1

**Table 2 sensors-22-06226-t002:** Details of the datasets.

Dataset	Class	Number	Color
Fabric Images	Hole	100	Red
	Thick bar	100	Green
	Thin bar	100	Yellow
	Broken end	100	Orange
	Knot	100	Purple
	Netting multiple	100	Cyan
	Non-defective	100	Black
TILDA	Hole	240	Red
	Color spot	240	Green
	Thread error	240	Yellow
	Foreign body	240	Orange
	Non-defective	240	Black

**Table 3 sensors-22-06226-t003:** Quantitative comparison of different detection methods.

Dataset	Metric	FCN	UNet	Mobile-Unet	LSTS	RPDNet
FI	IoU	0.645	0.687	0.727	0.650	**0.766**
	Recall	0.771	0.800	0.792	0.811	**0.833**
	Precision	0.798	0.822	0.898	0.766	**0.905**
	F1-Measure	0.784	0.809	0.842	0.788	**0.867**
TILDA	IoU	0.719	0.687	0.752		**0.831**
	Recall	0.831	0.784	0.833		**0.886**
	Precision	0.842	0.849	0.886		**0.930**
	F1-Measure	0.836	0.815	0.859		**0.907**

**Table 4 sensors-22-06226-t004:** Quantitative results of ablation experiments. w/o and w stand for without and with.

Method	IoU	Recall	Precision	F1-Measure
RPDNet w/o E2-D2 and E3-D3	0.767	0.850	0.887	0.868
RPDNet w blank input labels	0.801	0.866	0.914	0.890
RPDNet	**0.831**	**0.886**	**0.930**	**0.907**

**Table 5 sensors-22-06226-t005:** Results of extended experiments with RPDNet. We record the percentage improvement in parentheses.

Method	IoU	Recall	Precision	F1-Measure
UNet	0.687	0.784	0.849	0.815
RPDNet-UNet	0.730 (6.2%)	0.813 (3.7%)	0.877 (3.4%)	0.844 (3.6%)
MobileNetv2 and DeeplabV3+	0.759	0.819	0.911	0.863
RPDNet-MobileNetv2 and DeeplabV3+	0.820 (8.0%)	0.869 (6.1%)	0.935 (2.6%)	0.901 (4.4%)
GhostNet and DeeplabV3+	0.767	0.850	0.887	0.868
RPDNet-GhostNet and DeeplabV3+	0.831 (8.3%)	0.886 (4.2%)	0.930 (4.9%)	0.907 (4.5%)

**Table 6 sensors-22-06226-t006:** The experiment of proposed RPDNet on repeated-pattern-free images. w/o and w stand for without and with.

Method	IoU	Recall	Precision	F1-Measure
RPDNet w/o E2-D2 and E3-D3	0.754	0.825	0.897	0.860
RPDNet w blank input labels	**0.774**	0.833	**0.916**	**0.872**
RPDNet	0.772	**0.848**	0.895	0.871

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
