# Peer review of "RPDNet: Automatic Fabric Defect Detection Based on a Convolutional Neural Network and Repeated Pattern Analysis"

_sensors, 2022, doi:10.3390/s22166226_

Round 1

Author Response

Dear Reviewer:

Thank you for your letter and the comments concerning our manuscript entitled with “RPDNet: Automatic Fabric Defect Detection based on a Convolutional Neural Network and Repeated Pattern Analysis”. The comments are all valuable and helpful for improving the quality of our submitted paper. We have made the responses to the comments addressed by you and list these responses in an independent “Response letter”. A track copy is appended at the end of the response letter for your convenience to check every modification to the manuscript. 

If you have any question about this paper, please don’t hesitate to let me know. Thank you very much!

Sincerely yours

Reviewer 2 Report

 RPDNet: Automatic Fabric Defect Detection based on a Convolutional Neural Network and Repeated Pattern Analysis 

The authors have developed an RPDNet algorithm for detecting defects in fabrics using CNN. The database used in the study is TILDA which has been used by other authors in their work. The article is well written and well structured. The language of the paper is comprehendible. The scientific study is also designed well.

However, the authors have mentioned that the repeated pattern analysis, did not perform well in performing the designated task, which is the main novelty of the paper. If authors can introduce a small section on what could have possibly gone wrong in the architecture, it will help out future readers.

Rest, it is a wonderful article and should be published.

Some minor errors

1.         Line 28: With an accuracy

2.         Line 147: Mention the names of the authors from reference [35]

3.         Line 158: Methodology is wrongly numbered as 2

Author Response

(The authors gave the same response as above.)

Round 2

Reviewer 1 Report

The manuscript can be accepted as it is